# Fault Diagnosis Method Based on AUPLMD and RTSMWPE for a Reciprocating Compressor Valve

**DOI:** 10.3390/e24101480

**Published:** 2022-10-17

**Authors:** Meiping Song, Jindong Wang, Haiyang Zhao, Xulei Wang

**Affiliations:** 1Mechanical Science and Engineering Institute, Northeast Petroleum University, Daqing 163318, China; 2PetroChina Daqing Refining and Chemical Company, Daqing 163318, China

**Keywords:** adaptive uniform phase local mean decomposition, refined time-shift multiscale weighted permutation entropy, reciprocating compressor valve, feature extraction, fault diagnosis

## Abstract

In order to effectively extract the key feature information hidden in the original vibration signal, this paper proposes a fault feature extraction method combining adaptive uniform phase local mean decomposition (AUPLMD) and refined time-shift multiscale weighted permutation entropy (RTSMWPE). The proposed method focuses on two aspects: solving the serious modal aliasing problem of local mean decomposition (LMD) and the dependence of permutation entropy on the length of the original time series. First, by adding a sine wave with a uniform phase as a masking signal, adaptively selecting the amplitude of the added sine wave, the optimal decomposition result is screened by the orthogonality and the signal is reconstructed based on the kurtosis value to remove the signal noise. Secondly, in the RTSMWPE method, the fault feature extraction is realized by considering the signal amplitude information and replacing the traditional coarse-grained multi-scale method with a time-shifted multi-scale method. Finally, the proposed method is applied to the analysis of the experimental data of the reciprocating compressor valve; the analysis results demonstrate the effectiveness of the proposed method.

## 1. Introduction

Reciprocating compressors are the key equipment for compressing and transporting gas in petroleum, chemical, and other fields [1]. Once an accident occurs, it will cause huge economic losses and casualties. As a result of the complex structure, there are various reasons for failure, of which more than 60% will occur on the valve [2]. As the failure of mechanical equipment components is usually accompanied by the change in vibration signal, it is one of the more suitable methods to collect the vibration signal of the equipment and to make corresponding diagnosis and analysis. However, due to the complicated operation process of the reciprocating compressor, the extracted vibration signal is non-stationary and the extracted characteristic parameters are fuzzy, which brings great difficulty to the fault diagnosis of the reciprocating compressor [3]. In recent years, many fault diagnosis methods such as wavelet transform (WT), Wigner–Ville distribution (WVD), short-time Fourier transform (STFT), and other signal processing methods have been widely used and have achieved good results [4]. However, the above methods cannot effectively take into account the global and local characteristics of non-stationary signals in the time and frequency domains.

British scholar Jonathan S. Smith proposed an adaptive analysis method for non-stationary signals in 2005, namely local mean decomposition (LMD). LMD can adaptively decompose complex signals into the sum of several product functions (PF) and a residual component, where each PF component is obtained by multiplying an envelope signal and a pure FM signal in order to obtain the complete time–frequency distribution of the original signal [5,6]. Cheng et al. conducted a comparative study between LMD and empirical mode decomposition (EMD) methods, and the results show that LMD is superior to the EMD method in terms of the end effect and iteration number, and the decomposition accuracy is more ideal, and the instantaneous frequency of PF components is determined by a pure FM signal [7,8,9]. The problem of unexplained negative frequencies when the EMD is decomposed does not arise. However, similar to EMD, the LMD algorithm still suffers from some degree of modal aliasing and end effect issues. Inspired by EEMD, Yang et al. proposed an ensemble local mean decomposition (ELMD) method and pointed out that ELMD is more effective than EEMD in mechanical fault diagnosis. However, the ELMD method still suffers from the two intractable problems mentioned above [10,11,12]. Wang et al. applied the complementary integrated local mean decomposition CELMD method to the composite fault diagnosis of gearboxes, and effectively extracted the composite fault features, but the PF component was greatly affected by the amplitude of white noise. Lu et al. proposed an adaptive complementary ensemble local mean decomposition (ACELMD) method to decompose the marine underwater acoustic signal, reducing the modal aliasing phenomenon. Wang et al. proposed a complete ELMD with adaptive noise (CELMDAN), which further reduced residual noise, alleviated mode aliasing, and verified that CELMDAN outperformed CEEMDAN. Anh Ngoc-Lan Huynh et al. proposed the robust local mean decomposition (RLMD), which optimized the moving average algorithm and its screening process by adding adaptive amplitude white noise, and it achieved good results. The white noise used in auxiliary noise decomposition is a broadband signal with complex frequency components. In the process of mapping the signal to be decomposed to the corresponding frequency band of the white noise, modal aliasing appeared easily [13,14,15]. In 2018, Wang et al. reduced residual noise by adding the phase of the narrow-band sine wave, and proposed uniform phase empirical mode decomposition (UPEMD) [16], then Zheng et al. optimized the amplitude of the sine wave added by UPEMD [17]. The decomposition effect was found to be better than CEEMDAN. Therefore, inspired by its predecessors, this paper proposes adaptive uniform phase local mean decomposition (AUPLMD) with an optimization parameter. A new noise addition strategy was adopted to improve the computational efficiency and to alleviate the mode aliasing problem.

Entropy is a measure of system complexity. Depending on the type of failure and the degree of damage to the reciprocating compressor valve, its entropy value will also be different [18,19,20]. Permutation entropy (PE) is one of the most reliable and conceptually simple tools, and has been widely used in many fields [21,22,23]. However, the main drawback of this method is that it only considers the relative order structure of the time series, and ignores the amplitude information [24]. Fadlallah et al. proposed weighted permutation entropy (WPE) in 2013 to solve this problem [25,26,27]. WPE has a better anti-noise and anti-interference ability, its algorithm is simple, and its operability is strong. It can effectively amplify the small changes of the time series, and achieve good application results in mechanical fault diagnosis, medicine, and other fields [28,29]. However, WPE is only based on a single-scale measure. Therefore, a multiscale weighted permutation entropy (MWPE), which takes into account the complex temporal fluctuations inherent in the sequence, is proposed as an improved version of WPE [30,31,32]. However, it also exposes some shortcomings of MWPE. Because of the introduction of a weighting factor, small changes in the data series will lead to large fluctuations in MWPE. Moreover, the shortening of the time series in the coarse-grained process leads to a loss of information, which may lead to abrupt changes in MWPE and inaccurate estimates [33,34]. Therefore, this paper proposes refined time-shifted multiscale weighted permutation entropy (RTSMWPE) to measure the irregularity and complexity of the time series. In the RTSMWPE method, the coarse-grained-based multi-scale method used in MWPE is replaced by a new time-shifted multi-scale method, which has the following advantages. First, the method considers the amplitude information of the original time series to construct a new time-shifted multi-scale time series, which can effectively preserve the important structural information of the original data. Second, it reduces the dependence on data length and outperforms other similar methods when dealing with short time series.

In order to further reduce the interference of noise on feature extraction, a fault feature extraction method based on AUPLMD and RTSMWPE is proposed to improve the accuracy of fault diagnosis. In this paper, a masking signal with an adaptive sine wave amplitude is added to the LMD algorithm to find the optimal decomposition result and reconstruct the decomposition signal according to the value of kurtosis. Fault feature extraction is realized by the RTSMWPE method. The AUPLMD-RTSMWPE method is applied to the analysis of experimental data of the reciprocating compressor valves. By verifying the simulated vibration signals and the measured vibration signals, the AUPLMD method can effectively reduce the modal aliasing phenomenon. At the same time, compared with AUPLMD-MWPE, LMD-RTSMWPE, and LMD-MWPE, the method presented in this paper can extract early fault features accurately under a strong noise background, and has better practicability.

The rest of this paper is organized as follows. In Section 2, the AUPLMD signal decomposition method is introduced. In Section 3, the RTSMWPE feature extraction method is proposed. Section 4 verifies the effectiveness of the method in diagnosing the fault of reciprocating compressor valve.

## 2. Adaptive Uniform Phase Local Mean Decomposition

### 2.1. Local Mean Decomposition

LMD is a novel time–frequency analysis method that adaptively decomposes complex signals into PF components, in which multiple envelope signals are multiplied by pure frequency modulated signals. The construction of the mean function and envelope estimation in the traditional LMD method is obtained by smoothing the local mean line segment and the local amplitude line segment using the moving average method. However, the subjectivity of the moving average step size selection and the phase error may occur in the multiple smoothing process, which are related to the LMD decomposition accuracy. Therefore, a monotonic piecewise cubic Hermite interpolation is proposed to replace the moving average method in order to solve the problems in envelope construction. For the signal *x*(*t*), the LMD decomposition process is as follows.

Extract the array of the local extreme points (maximum points and minimum points) of the original signal *x*(*t*), and use the monotonic piecewise cubic Hermite to interpolate it to construct the upper envelope function *env_max_*(*t*) and the lower envelope function *env_min_*(*t*). Calculate the local mean function *m*_11_(*t*) and the envelope estimation function *a*_11_(*t*) using the upper and lower envelopes:(1)$$m_{11}=\frac{env_{\max}\left(t\right)+env_{\min}\left(t\right)}{2}$$
(2)$$a_{11}=\frac{\left|env_{\max}\left(t\right)-env_{\min}\left(t\right)\right|}{2}$$

Separate *m*_11_(*t*) from signal *x*(*t*) to obtain another signal $h_{11}\left(t\right)=x\left(t\right)-m_{11}\left(t\right)$, and the frequency modulation signal $s_{11}\left(t\right)=h_{11}\left(t\right)/a_{11}\left(t\right)$ is obtained.

Detect *s*_11_(*t*) to judge whether it is a pure FM signal. If $a_{12}\left(t\right)=1$, then *s*_11_(*t*) is a pure FM signal; if $a_{12}\left(t\right)\neq 1$, *s*_11_(*t*) is used as the original signal to repeat the above iterative process until *s*_1*n*_(*t*) becomes a pure FM signal, that is the envelope satisfies *a*_1(*n*+1)_ = 1, and the termination condition is $\underset{n\to \infty }{\lim}a_{1n}\left(t\right)=1$.

The envelope estimate signal *a*_1*q*_(*t*) are multiplied together to obtain the envelope signal of the first *PF* component.
(3)$$a_{1}\left(t\right)=a_{11}\left(t\right)a_{12}\left(t\right)\cdots a_{1n}\left(t\right)=\prod_{q-1}^{n}a_{1q}\left(t\right)$$

The first *PF* component *PF*_1_(*t*) is defined as
(4)$$PF_{1}=a_{1}\left(t\right)s_{1n}\left(t\right)$$

*PF*_1_(*t*) is separated from the original signal *x*(*t*) to obtain a new signal *u*_1_(*t*). The above process is repeated *k* times with *u*_1_(*t*) as the original signal until *u_k_*(*t*) is a monotonic function. The iterative equation is
(5)$$u_{k}\left(t\right)=u_{k-1}\left(t\right)-PF_{k}\left(t\right)$$

Finally, the original signal *x*(*t*) is expressed as
(6)$$x\left(t\right)=\sum_{p=1}^{k}PF_{k}\left(t\right)+u_{k}\left(t\right)$$

### 2.2. Uniform Phase Local Mean Decomposition

UPLMD homogenizes the distribution of extreme points by adding a narrow-band cosine signal with a uniform phase change to the signal to be decomposed, and achieves the purpose of suppressing modal aliasing. After removing the narrow-wave cosine signal in the decomposition result, the integrated averaging can eliminate the residual of the auxiliary signal, and increasing the phase number of the cosine wave can better suppress the modal aliasing and spurious components. For the vibration signal *x*(*t*), the decomposition process is as follows:

Calculate the number of loops *n_imf_* = log_2_(*n*) and the period *T_w_* according to the data length *n*:(7)$$T_{w}=2^{m},m=1:n_{imf}$$

The frequency *f_w_* = 1/*T_w_* of the cosine signal is obtained according to the period, and then the number of phases *n_p_* and the amplitude *ε* are set according to the actual signal *x*(*t*).

The phase *θ_k_* is evenly divided into equal parts in the range of [0, 2π], and the number of phases *np* must be the integer power of 2, and different amplitudes should be selected for different signals.

Let *r*_0_(*t*) = *x*(*t*), construct the narrow wave cosine signal *w*(*t*;*ε_m_*;*f_w_*;*θ_k_*), which is as follows:(8)$$w\left(t;\varepsilon_{m};f_{w};\theta_{k}\right)=\varepsilon_{m}\cdot \cos\left(2\pi \cdot f_{w}\cdot t+\theta_{k}\right)$$
where *ε_m_* is the product of the amplitude *ε* and the standard deviation value of *r_m_*_−1_(*t*); *θ_k_* is defined as *θ_k_* = 2π(*k* − 1)/*n_p_*.

The operator *L_i_*(·) is defined as the *i*-th PF decomposed by LMD, and use the LMD to decompose the signal after adding the narrow wave to obtain the first component:(9)$$c_{m,k}\left(t\right)=L_{1}\left(x\left(t\right)+w\left(t;\varepsilon_{m};f_{w};\theta_{k}\right)\right)k=1,2,\cdots ,n_{p};m=1:n_{imf}$$

After subtracting the narrow wave cosine signal *w*(*t*;*ε_m_*;*f_w_*;*θ_k_*) from *c_m_*_,*k*_(*t*), the *PF*_1_ of UPLMD is obtained through averaging, as follows
(10)$$PF_{1}=\left(\sum_{k=1}^{n_{p}}c_{m,k}\left(t\right)-w\left(t;\varepsilon_{m};f_{w};\theta_{k}\right)\right)/n_{p}$$

Separate the *PF*_1_ component from the original signal *x*(*t*), and use the remaining part *r_i_*(*t*) as a new signal, repeat the above steps until all *PF_m_* (*m* = 2,3, …, log_2_*n*) components are decomposed, $r_{\log_{2}n}\left(t\right)$ is a trend term.

### 2.3. Adaptive Uniform Phase Local Mean Decomposition

When UPLMD is used to process the collected vibration signal, its two important parameters, the number of phases *n_p_* and the amplitude *ε* of narrow wave cosine signal, need to be artificially selected in advance. Thus, it does not possess an adaptive capability. The value of the phase number *n_p_* directly affects the decomposition ability of UPLMD. The larger the *n_p_*, the higher the decomposition ability and the less noise residue, but the calculation time will be longer. Usually, *n_p_* is in the range of 4~32 to balance the decomposition ability and time cost, so the value of *n_p_* is 16. For the selection of the amplitude value, different signals have different optimal amplitudes, so it is difficult to choose empirically. Therefore, AUPLMD is adopted to automatically select the important parameters of UPLMD. Through massive experiments and analysis, the amplitude *ε_m_* is selected in the range of 0.10~0.5, where 0.02 is the step size, and the orthogonality index of the decomposition result is compared and the optimal amplitude is automatically selected within the range. Orthogonality is an important index to measure modal aliasing. In theory, the orthogonality index for single-component *PF* is 0, otherwise it will increase. Therefore, the optimal *PF* criterion is adaptively selected based on the minimum orthogonality in order to improve the decomposition ability and accuracy of AUPLMD. The flow chart of AUPLMD is shown in Figure 1, and the specific steps of AUPLMD are introduced as follows: Step 1:Set the number of phases as *n_p_* = 16, then adaptively select the optimal amplitude *ε_mo_* in the given range and construct the masking signal *w*(*t*;*ε_mo_*;*f_w_*;*θ_k_*).Step 2:Decompose $x\left(t\right)+w\left(t;\varepsilon_{mo};f_{w};\theta_{k}\right)$ by LMD to obtain the first component $c_{m,k}\left(t\right)$, subtracting the narrow wave cosine signal *w*(*t*;*ε_mo_*;*f_w_*;*θ_k_*) from *c_m_*_,*k*_(*t*), the first component *PF*_1*j*_ is obtained through averaging as $PF_{1j}=\left(\sum_{k=1}^{n_{p}}c_{m,k}\left(t\right)-w\left(t;\varepsilon_{m};f_{w};\theta_{k}\right)\right)/n_{p}$.Step 3:Let *j* = *j* + 1, implementing step (2) until *ε_m_* takes different values in the range of 0.10~0.5 and it obtains a series of *PF*_1*j*_ components. The orthogonality index is used as the criterion to select the optimal *PF*_1_ component. The smaller the value of OI, the better the decomposition performance.Step 4:Separate the *PF*_1_ component from the original signal *x*(*t*), and use the remaining part *r_i_*(*t*) as a new signal. Then, repeat steps 1 to 3 until *x*(*t*) is finally decomposed into the sum of the *PFs* and a trend item.

### 2.4. Comparison Analysis

#### 2.4.1. Simulation Signal Analysis

To demonstrate the excellent decomposition performance of the proposed AUPLMD method, the decomposition results of a simulation signal (signal *S*_4_) constructed from three typical mechanical vibration signals are compared using LMD, CELMDAN, and the proposed AUPLMD. The simulation signal is shown in (11). Specifically, *S*_1_ is a cosine signal with frequency *f*_1_, *S*_2_ is a frequency modulated signal with a carrier frequency *f*_2_ and a modulation frequency *f_m_*, and *S*_3_ is an impulse signal [35]. *S*_4_ was a mixed signal consisting of these three signals, with a sampling frequency *f* = 1000 Hz and the number of sample points *N* = 2000. The time domain waveform and spectrums of *S*_4_ are shown in Figure 2.
(11)$$\left\{ \begin{array}{l} S_{1}=\sin\left(2\pi f_{1}t\right)\\ S_{2}=\sin\left(2\pi f_{2}t+\cos\left(2\pi f_{m}t\right)\right)\\ S_{3}=e^{-10t}*\sin\left(2\pi f_{3}t\right)\\ S_{4}=S_{1}+S_{2}+S_{3} \end{array}\right.$$
where *f*_1_ = 10 Hz, *f*_2_ = 400 Hz, *f*_3_ = 800 Hz, and *f_m_* = 25 Hz.

Signal *S*_4_ is decomposed by three methods, namely LMD, CELMDAN, and AUPLMD, as shown in Figure 3. The decomposition results of LMD show serious modal aliasing. Compared with the LMD method, the CELMDAN method still has modal aliasing. Compared with the other two methods, the decomposition result of AUPLMD is the best, and its performance is very encouraging, because its decomposition results show only slight mode aliasing. *PF*_1_, *PF*_2_, and *PF*_6_ in the AUPLMD decomposition result are highly consistent with signal *S*_2_, signal *S*_3_, and signal *S*_1_, respectively (Figure 2), which indicates that AUPLMD can effectively reduce modal aliasing.

#### 2.4.2. Valve Analog Signal Analysis

The advantages of the AUPLMD method are verified by the example of a normal working state of valve. Vibration signals of the valve in a normal working state containing two full-cycle data points are selected, as shown in Figure 4, and signal decomposition is performed using the above methods, as shown in Figure 5. It can be seen that the decomposition ability of LMD is poor, and the mode aliasing phenomenon of the *PF*_2_ component is the most serious. Although CELMDAN is superior to LMD, modal aliasing also appears, such as the *PF*_4_ component. However, the *PF*_1_–*PF*_4_ components in CELMDAN are decomposed into *PF*_1_–*PF*_6_ components in AUPLMD, and the modal aliasing phenomenon among each component is alleviated, which indicates that AUPLMD also has a good effect on vibration signal decomposition under actual working conditions, and the mode aliasing can be effectively reduced.

## 3. Refined Time-Shifted Multiscale Weighted Permutation Entropy

### 3.1. Multiscale Weighted Permutation Entropy

PE is an effective method to measure the complexity of the nonlinear time series proposed by Bandt and Pompe. For a given time series, *X* = {*x*(*n*), *n* = 1, 2, …, *N*}, with the length being *N*, the phase space reconstruction of the time series is obtained $X_{K}=\left\{x\left(i\right),x\left(i+d\right),...,x\left(i+\left(m-1\right)d\right)\right\},$i=1,2,...,K, where K=N−(m−1)d, *m* is the embedding dimension, and *d* is the time delay. Arrange each vector in the reconstructed time series $X_{K}$ in ascending order and obtain $x\left(i+\left(j_{1}\left(i\right)-1\right)d\right)\leq$$x\left(i+\left(j_{2}\left(i\right)-1\right)d\right)\leq \cdots$ $\leq x\left(i+\left(j_{m}\left(i\right)-1\right)d\right)$, let $\pi_{i}=\left(j_{1}\left(i\right),j_{2}\left(i\right),\cdots ,j_{m}\left(i\right)\right),$i=1,2,⋯,K represents the index of the column in the reconstruction component where the element is located, and *π_i_* is a certain arrangement of (1,2,···,*m*). Obviously, m elements have at most m! different arrangements, denoted as Π. Count all of the permutations of *π_i_*, find out the frequency of occurrence of each permutation $\pi_{r}\left(0<r\leq m!\right)$, namely:(12)$$p\left(\pi_{r}\right)=\frac{\sum_{j=1}^{K}1_{u:type\left(u\right)=\pi_{r}}\left(X_{j}\right)}{\sum_{j=1}^{K}1_{u:type\left(u\right)\in \prod }\left(X_{j}\right)}$$
where $1_{u}\left(v\right)=\left\{ \begin{array}{l} 1,v\in u\\ 0,v\notin u \end{array}\right.,\prod ={\left\{\pi_{r}\right\}}_{r=1}^{m!}$.Considering the amplitude information of the time series, there is an amplitude difference between the sequences with the same arrangement, so when calculating the frequency of each arrangement with the corresponding weight, the frequency of each arrangement pattern is defined as follows:(13)$$p\left(\omega \pi_{r}\right)=\frac{\sum_{j=1}^{K}1_{u:type\left(u\right)=\pi_{r}}\left(X_{j}\right)\omega_{r}}{\sum_{j=1}^{K}1_{u:type\left(u\right)\in \prod }\left(X_{j}\right)\omega_{r}}$$
where $\omega_{r}$ is the weight of the reconstructed component *X_j_*, and it is represented by the variance of *X_j_*:$\omega_{r}=\frac{1}{m}\sum_{q=1}^{m}{\left[x\left(j+\left(q-1\right)d\right)-{\bar{X}}_{j}\right]}^{2}$, ${\bar{X}}_{j}$ is the mean of the reconstructed component *X_j_*: ${\bar{X}}_{j}=\frac{1}{m}\sum_{q=1}^{m}x\left(j+\left(q-1\right)d\right)$.

WPE is an improved algorithm of permutation entropy, which considers the amplitude information contained in the time series, and the expression is defined as follows:(14)$$\mathrm{WPE}\left(x,m,d\right)=-\sum_{r=1}^{m!}p\left(\omega \pi_{r}\right)\ln\left(p\left(\omega \pi_{r}\right)\right)$$

The calculation process of MWPE mainly includes two steps:

First, coarse-grained processing is performed on the original time series *X*. According to the scale factor *τ*, divide *X* into *N*/*τ* non-overlapping segments, each segment contains *τ* data points, calculate the arithmetic mean of *τ* data points in each segment to represent the value of this segment, and the coarse-grained time series is defined as follows:(15)$$y_{k,j}^{\tau }=\frac{1}{\tau }\sum_{i=\left(j-1\right)\tau +k}^{j\tau +k-1}x\left(i\right),1\leq j\leq \frac{N}{\tau },\;1\leq k\leq \tau$$

Second, calculate the MWPE value of the coarse-grained time series $y_{k,j}^{\tau }$ in the scale factor *τ*. The entropy value of the coarse-grained series when *τ* = 1 is the value obtained by the WPE method, the constraint N/τ>>m! must be satisfied in order to gain reliable statistics.
(16)$$\mathrm{MWPE}\left(x,m,d,\tau \right)=-\sum_{r=1}^{m!}p^{\tau ,k}\left(\omega \pi_{r}\right)\ln\left(p^{\tau ,k}\left(\omega \pi_{r}\right)\right)$$

### 3.2. Refined Time-Shift Multiscale Weighted Permutation Entropy

The insufficient coarse-graining process of MWPE makes the obtained coarse-grained time series greatly reduce the information richness of the original time series. In order to solve the excessive dependence of the coarse-grained time series on the length of the original time series in the MWPE algorithm, this paper proposes the RTSMWPE algorithm, and the detailed steps of RTSMWPE can be described as follows:

First, for a given original time series {*x*(*i*), *i* = 1,2, …, *N*}, define $X_{k}^{\beta }$ by
(17)$$X_{k}^{\beta }=\left(x_{\beta },x_{\beta +k},x_{\beta +2k},\cdots ,x_{\beta +n_{\beta }k}\right)$$
where *k* (*k* = *τ*) and *β* (1 ≤ *β* ≤ *k*) are positive integers, representing the starting point and interval time of the time series, respectively. *n_β_* is a rounded integer, indicating the number of upper boundaries, *n_β_* = (*N* − *β*)/*k*. For convenience, *k* is still called the scale factor. 

Second, for a given scale factor *k*, $X_{k}^{\beta }$ can obtain *k* new time series consisting of *k* time shift from the *β*th (*β* = 1,2, …, *k*) data point, and thus the $p_{k}^{\beta }\left(\omega \pi_{r}\right)$ of each $X_{k}^{\beta }$ can be obtained according to WPE’s computation. Then, define $\bar{p}\left(\omega \pi_{r}\right)=\frac{1}{k}\sum_{1}^{k}p_{k}^{\beta }\left(\omega \pi_{r}\right)$.

Last, the RTSMWPE of *X* = {*x*(*i*), *i* = 1,2, …, *N*} can be defined as
(18)$$RTSMWPE\left(X,m,d,\tau \right)=-\sum_{r=1}^{m!}\bar{p}\left(\omega \pi_{r}\right)\cdot \ln\left(\bar{p}\left(\omega \pi_{r}\right)\right)$$

Compared with MWPE, RTSMWPE avoids the phenomenon of amplitude “neutralization” in the coarse-grained process, and the calculation process of RTSMNDE is shown in Figure 6.

### 3.3. Comparison Analysis

In this section, a widely used simulation signal with white Gaussian noises (WGN) is used to analyze the proposed RTSMWPE method and it is compared with multiscale permutation entropy (MPE), multiscale weighted permutation entropy (MWPE), composite multiscale weighted permutation entropy (CMWPE), time-shift multiscale weighted permutation entropy (TSMWPE), and refined composite multiscale weighted permutation entropy (RCMWPE) to verify its effectiveness. The lengths of the time series of white noise are 2048, 4096, 6144, 8192, and 10,240. The entropy results are shown in Figure 7a–f. The calculation of RTSMWPE is related with the following parameters: embedding dimension m and time delay d. According to the authors of [36], choose *m* = 5, *d* = 1, and scale factor *τ* = 20.

It can be seen from Figure 7a,b that when the scale factor *τ* = 1, the entropy value is the largest, close to 1. As the scale factor increases, the entropy value gradually decreases, while under the same scale factor, the entropy value increases with the length of the time series. However, compared with MPE, the MWPE curve exhibits small fluctuations, which is because the MWPE takes into account the amplitude information of the time series. At the same time, under the same scale factor and data points, the entropy obtained by MWPE is lower than that of MPE. Figure 7c,d shows that with the increase in the scale factor, the change trend of the entropy curves of CMWPE and TSMWPE is the same as the previous two methods, but the CMWPE and TSMWPE curves are more stable. Figure 7e,f indicates that the entropy curves of RCMWPE and RTSMWPE remain stable as the scale factor increases. However, when the time series length is *N* = 2048, the entropy value of RCMWPE decreases slowly with the increase in the scale factor, while the entropy value of RTSMWPE remains basically unchanged, and there are small fluctuations. From the above analysis, it can be concluded that the RTSMWPE method proposed in this paper has strong independence on the length of the time series.

In order to verify the merits of RTSMWPE in stability, 100 groups of white Gaussian noise as sample signals with a length of 2048 were selected for stability research, and the results are shown in Figure 8. By comparing this with MPE, MWPE, CMWPE, TSMWPE, and RCMWPE, two conclusions can be drawn. First, the proposed RTSMWPE curve is smoother and more stable than the other five methods. Second, the RTSMWPE method obtained smaller error bars, especially at larger scales. In conclusion, the RTSMWPE method has the advantage of stability.

## 4. Fault Diagnosis Method of Reciprocating Compressor Valve Based on AUPLMD and RTSMWPE

### 4.1. Proposed Method

This paper proposes a new feature extraction method based on AUPLMD and RTSMWPE, and applies it to the fault diagnosis of reciprocating compressor valves. The fault diagnosis flowchart is shown in Figure 9. The specific steps are as follows:Step 1:The data acquisition under different health conditions is conducted;Step 2:The AUPLMD method is used to adaptively decompose the vibration signal of the valve under different fault states into nimf number of *PF*s, and use the kurtosis criterion to select the *PF* component that can significantly represent the fault characteristics and reconstruct the fault signal;Step 3:Quantization processing of the four reconstructed signals, the RTSMWPE values of the four reconstructed signals were solved separately in order to obtain the four fault eigenvectors of the reciprocating compressor valve.Step 4:Segment the data into 100 samples for each health condition and divide the obtained samples into the training set and testing set. Use a support vector machine (SVM) to train and test the valve fault feature vector to identify different fault types and sub-health conditions.

### 4.2. Experimental Data Analysis

Reciprocating compressors have been widely used in the petroleum and chemical industries, and their operational status and safety are considered challenging research topics. In this study, the proposed AUPLMD and RTSMWPE methods were used to extract fault features from valve vibration data in the 2D12-70/0-13 type two-stage double-acting reciprocating compressor, as shown in Figure 10. Its main design parameters are as follows: shaft power is 500 kW, piston stroke is 240 mm, and crankshaft speed is 496 rpm.

Reciprocating compressor valves are often subjected to long-term alternating loads and the valves are prone to failure. When the valve is abnormal, the performance of the vibration signal in the direction of the bonnet side will change greatly. Therefore, this paper extracts the bonnet vibration signal as the analysis data. The locations of the sensors are marked with red squares in Figure 10a. To collect vibration signals, tests were performed using a Hubei UT3416 data acquisition instrument and an integrated circuit piezoelectric (ICP) accelerometer placed on top of the valve surface. The sensor sensitivity is 100 mv/g, the measurement range is ±50 g, and the frequency range is 0.5–5 kHz.

In this paper, the failure state test of the valve on the secondary valve cover side was carried out under actual working conditions, and four types of reciprocating compressor valve states were mainly studied: normal state, spring failure state (SFS), valve plate fracture (VPF), and valve plate gap (VPG). The valve is shown in Figure 11, Figure 11a is the reciprocating compressor valve, Figure 11b is the valve plate under normal working conditions, Figure 11c is the valve plate with a gap on the second rings of the outer ring number states, Figure 11d shows the condition of one less spring and the red box is the missing spring position, and Figure 11e shows the middle valve plate broken. Two periods of vibration signals for the four main valve states are shown in Figure 12.

In this paper, the AUPLMD method is used to decompose the four state signals of the reciprocating compressor. According to the signal length, AUPLMD decomposes the signal into 13 *PFs*. Calculate the kurtosis value of the *PF* component according to the kurtosis criterion, as shown in Table 1. If the signal contains more fault components, the kurtosis value of the signal will be larger, so the *PF* component with a larger absolute value of kurtosis should be selected to reconstruct the signal, so as to analyze the signal of each state. In this paper, *PF* components with a kurtosis value greater than 6 were selected for signal reconstruction. The reconstructed signals of the four valve states are shown in Figure 13.

Next, calculate the RTSMWPE values of the original vibration signal and the reconstructed vibration signal under different valve states, where *m* = 5, *d* = 1, and the maximum scale factor is 20, as shown in Figure 14. For the reconstructed signal, with the increase in the scale factor, the entropy values of the four valve states increases, but the entropy values fluctuate greatly. When the scale factor is greater than 8, the entropy values of the four valve states tend to be stable. Moreover, compared with the original vibration signal, the vibration signal decomposed and reconstructed by AUPLMD has a better distinguishability. In summary, in order to have a better fault diagnosis effect, select the entropy value of 12 scales with *τ* > 8 as the state fault eigenvector.

### 4.3. Discussion and Recommendations

For the experimental data of the above four valves with different fault states, each state contains 100 samples, and a total of 400 samples are obtained. Calculate the RTSMWPE values for all 400 samples, where *m* = 5 and *d* = 1. Here, 60 training samples and 40 testing samples are randomly selected from the 100 samples of each state. The training samples and test samples are input into the SVM for training and test identification successively, and the output results are shown in Figure 15. The failure identification rate of the spring failure state is the highest (97.5%), and the failure identification rate of valve plate gap is the lowest (90%). The overall diagnosis rate based on AUPLMD–RTSMWPE–SVM in this article is as high as 95%.

In order to further verify the superiority of this method, the same valve faults for the data samples are diagnosed by combining the multi-fault classifier by SVM using AUPLMD–MWPE, LMD–RTSMWPE, and LMD–MWPE, respectively. The specific diagnosis result is shown in Table 2. From the identification results, it can be seen that the total accuracy of the AUPLMD–MWPE, LMD–RTSMWPE, and LMD–MWPE methods to identify using SVM is 89.375%, 91.875%, and 87.5%, respectively, which is lower than the method proposed in this paper. Furthermore, the accuracy of the three methods for single valve failure is lower than AUPLMD–RTSMWPE. The LMD–MWPE method without any improvement has the lowest fault accuracy. Through the above analysis, the effectiveness of the AUPLMD–RTSMWPE fault diagnosis method of reciprocating compressor valves is verified.

## 5. Conclusions

Aiming at the characteristics of the vibration signal of the reciprocating compressor valve, a fault diagnosis method based on AUPLMD and RTSMWPE is proposed and applied to the fault diagnosis of the reciprocating compressor valve.

This method proposes a uniform phase local mode decomposition method that adds a sine wave to the signal to be decomposed, and optimizes the amplitude of the auxiliary signal. Compared with the traditional LMD decomposition method, our method achieves a good effect for suppressing modal aliasing.

A refined time-shifted multi-scale weighted permutation entropy method is proposed to extract features from the signal, which not only considers the relative order structure of the time series, but also considers the amplitude information of the signal. By constructing time-shifted multi-scale time series, it reduces the dependence on the data length and outperforms other similar methods when dealing with a short time series.

The simulation analysis shows that the method proposed in this paper has the best fault diagnosis ability compared with other methods for the fault diagnosis of the reciprocating compressor valve.

The proposed method has some limitations. First, the parameter optimization process of the AUPLMD method takes a long time. Second, the feature extraction method based on permutation entropy is lower than other entropy methods for the accuracy of fault diagnosis. In future work, we will focus on the research of feature extraction methods.

## Figures and Tables

**Figure 1 entropy-24-01480-f001:**
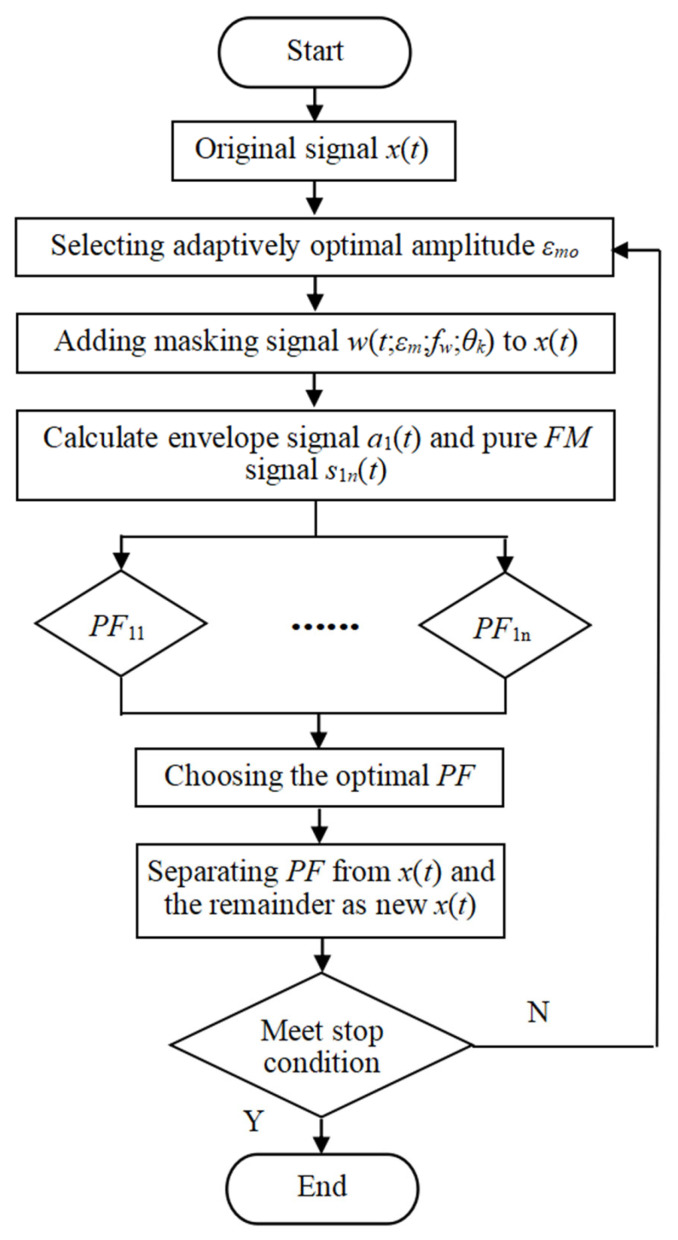
The flow chart of AUPLMD.

**Figure 2 entropy-24-01480-f002:**
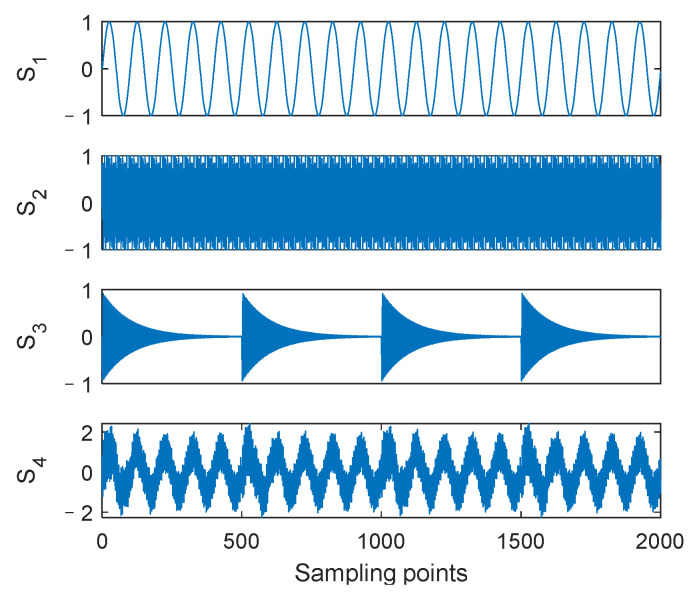
Waveforms of the four simulated signals.

**Figure 3 entropy-24-01480-f003:**
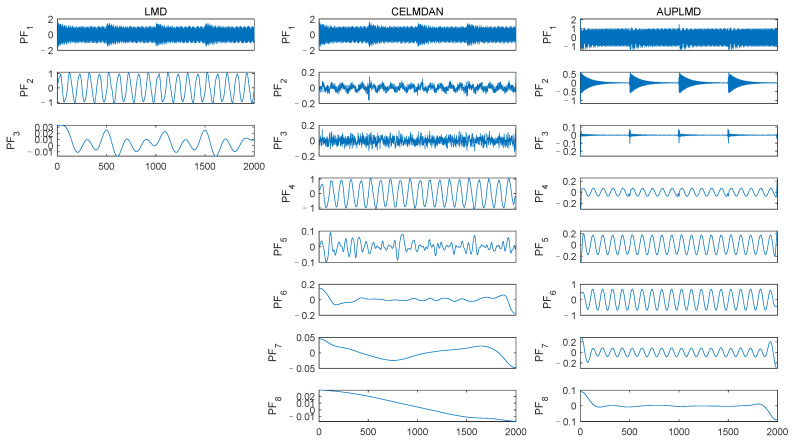
Decomposition results of the simulated signal *S*_4_ using LMD, CELMDAN, and AUPLMD.

**Figure 4 entropy-24-01480-f004:**
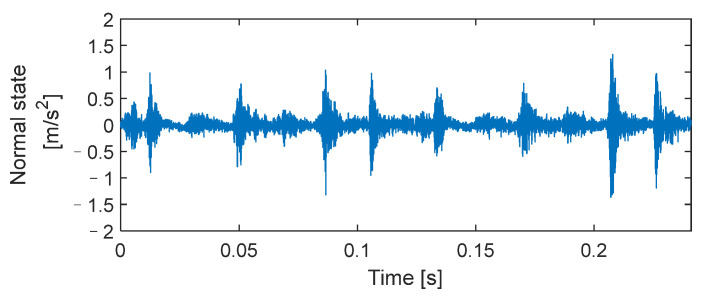
Vibration signal of a reciprocating compressor valve in a normal working state.

**Figure 5 entropy-24-01480-f005:**
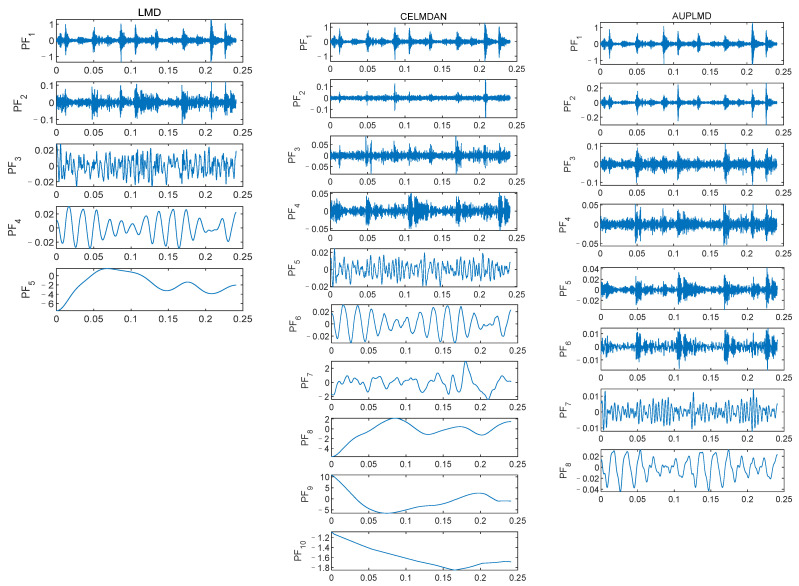
Decomposition results of the valve analog signal using LMD, CELMDAN, and AUPLMD.

**Figure 6 entropy-24-01480-f006:**
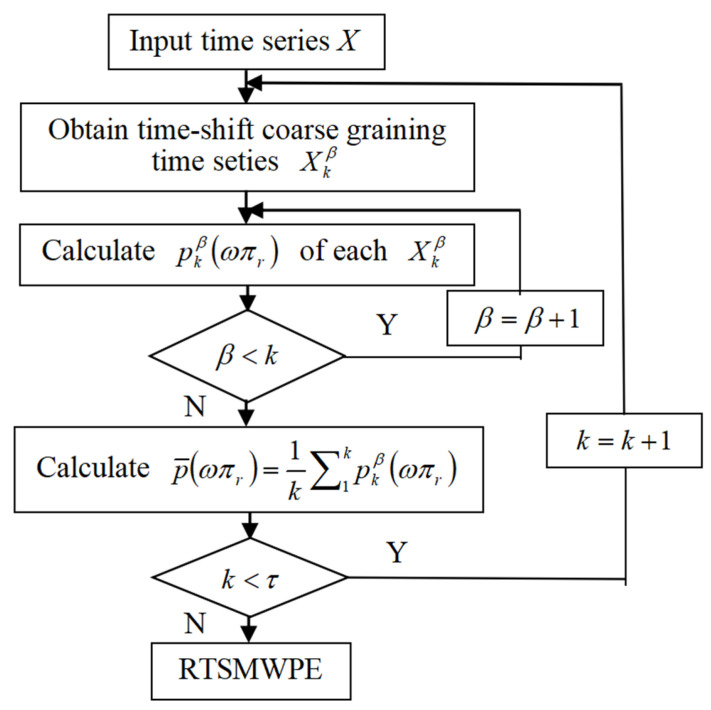
Calculation process of RTSMWPE.

**Figure 7 entropy-24-01480-f007:**
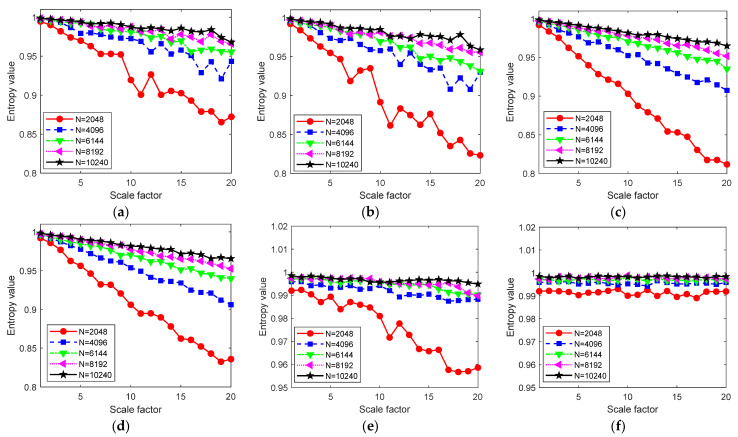
Comparison of entropy values obtained under different lengths of WGN: (**a**) MPE, (**b**) MWPE, (**c**) CMWPE, (**d**) TSMWPE, (**e**) RCMWPE, and (**f**) RTSMWPE.

**Figure 8 entropy-24-01480-f008:**
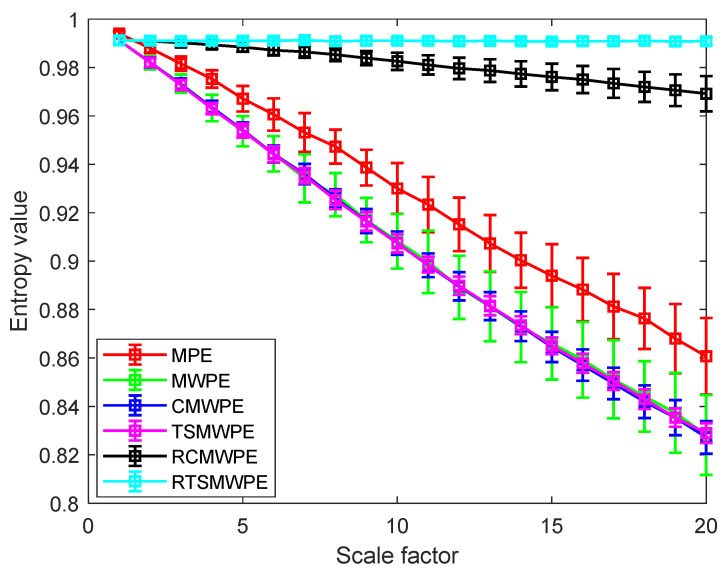
Comparison analysis of MPE, MWPE, CMWPE, TSMWPE, RCMWPE, and RTSMWPE.

**Figure 9 entropy-24-01480-f009:**
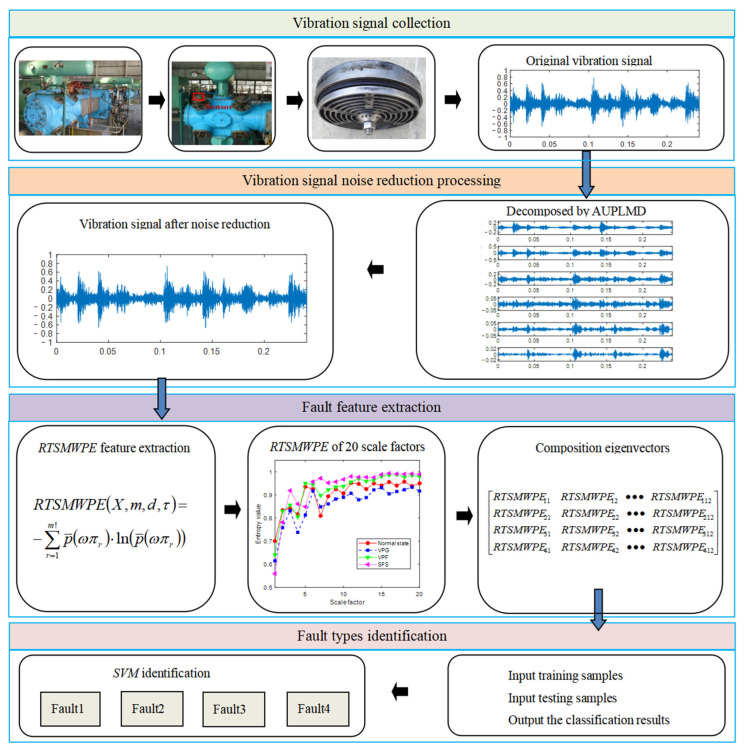
Flow chart of fault diagnosis.

**Figure 10 entropy-24-01480-f010:**
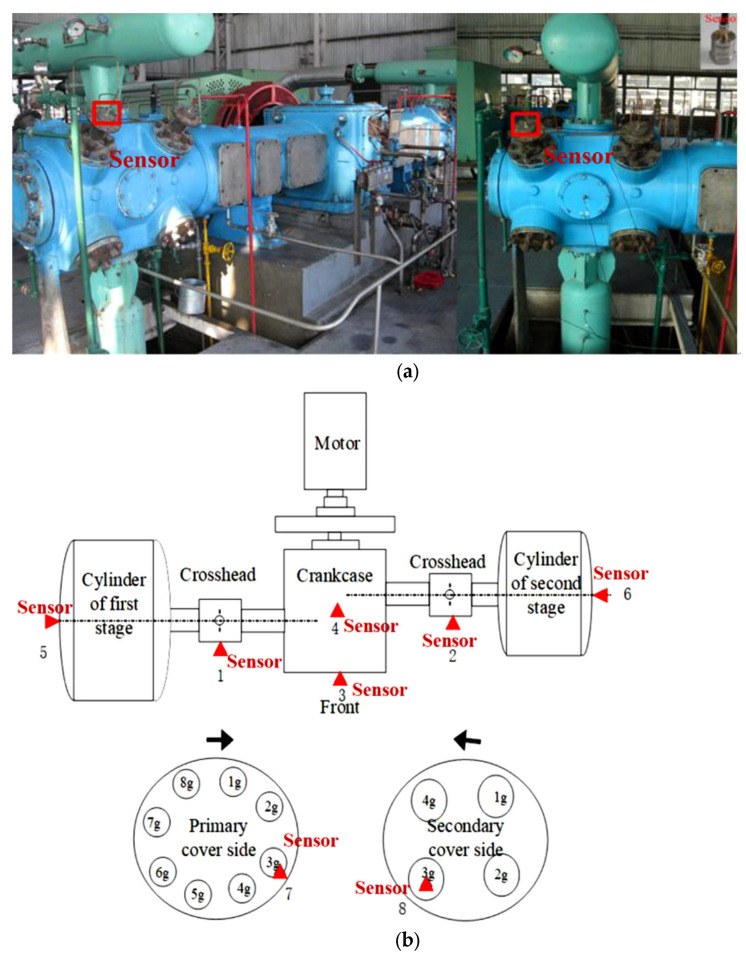
Two-stage double-acting reciprocating compressor of 2D12 type; (**a**) test bench of the reciprocating compressor, and (**b**) structural drawing of reciprocating compressor and measuring points.

**Figure 11 entropy-24-01480-f011:**
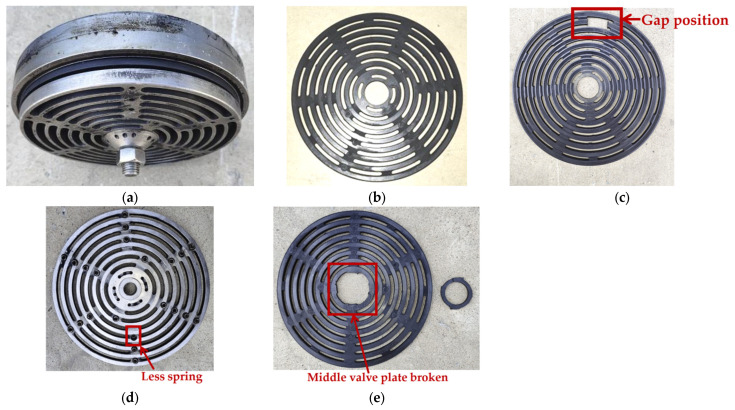
Reciprocating compressor valve: (**a**) valve, (**b**) valve plate, (**c**) valve plate gap, (**d**) one less spring, and (**e**) the middle valve piece breaks.

**Figure 12 entropy-24-01480-f012:**
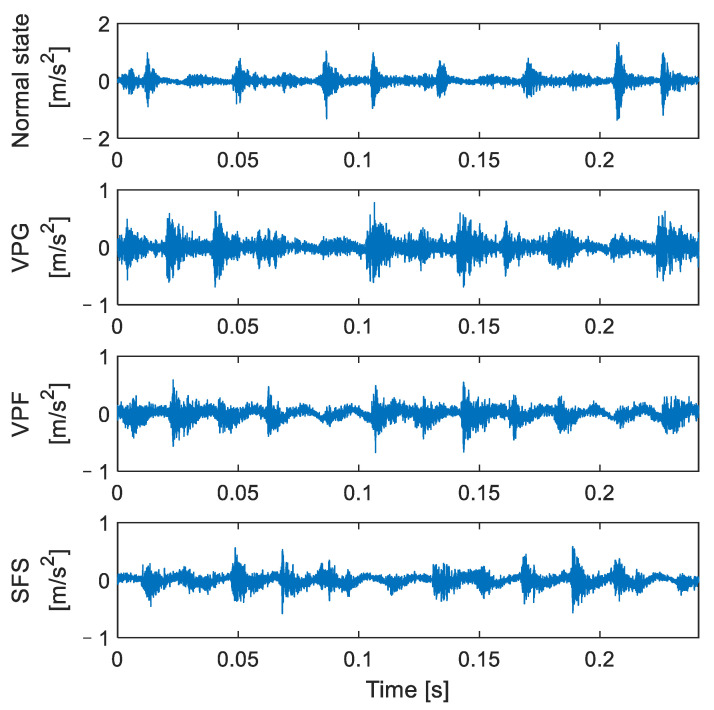
Time-domain waveform diagram of reciprocating compressor under four valve states.

**Figure 13 entropy-24-01480-f013:**
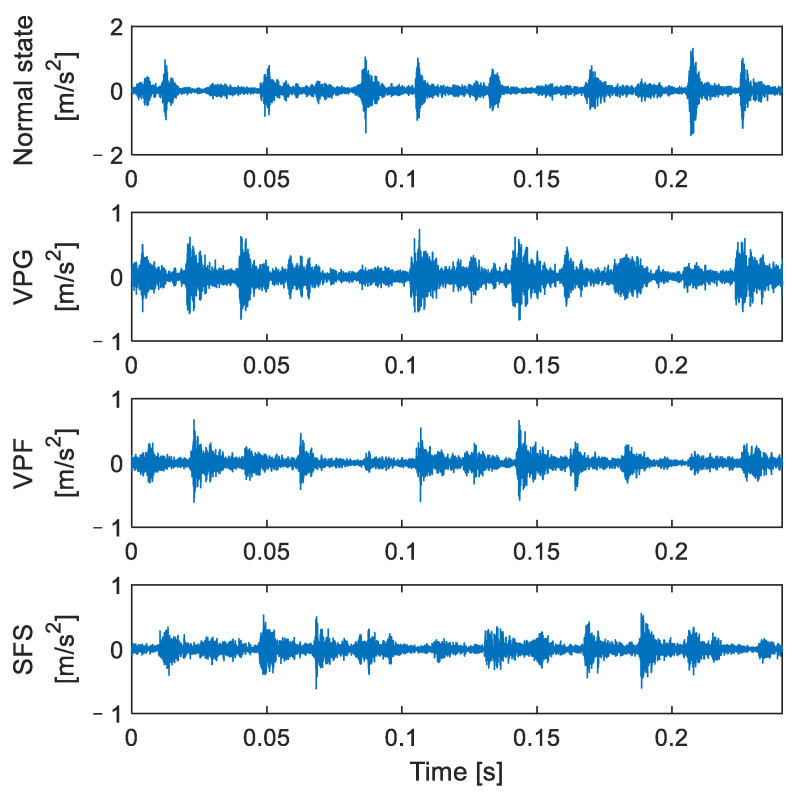
Valve reconstruction signal.

**Figure 14 entropy-24-01480-f014:**
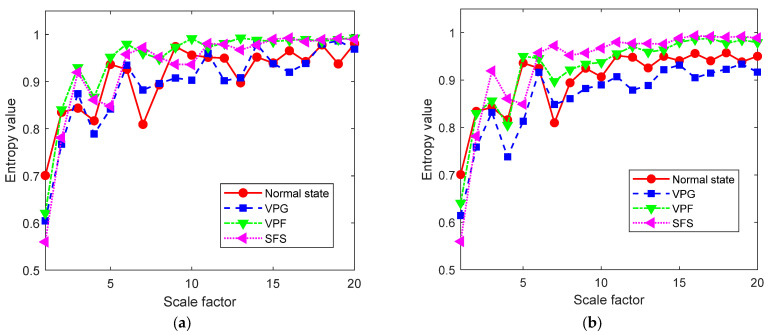
The RTSMWPE curve diagram of valve in four states: (**a**) RTSMWPE value of the original vibration signal, and (**b**) RTSMWPE value of the reconstructed vibration signal.

**Figure 15 entropy-24-01480-f015:**
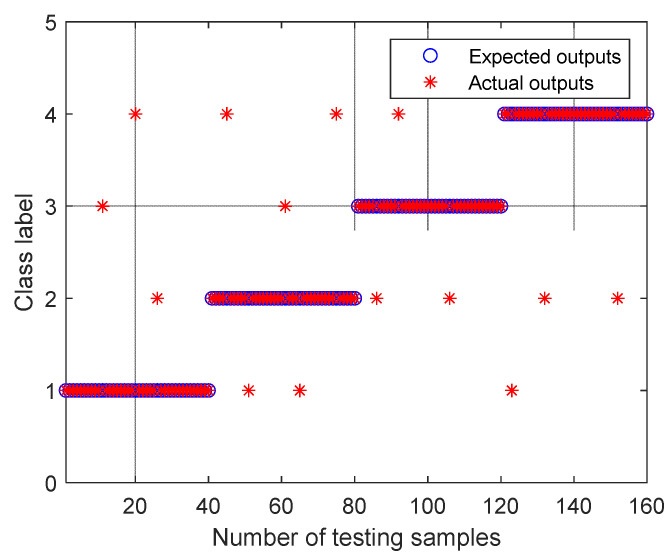
Outputs of the proposed method for all of the testing samples.

**Table 1 entropy-24-01480-t001:** The kurtosis values of each PF component under different valve states.

Valve States	The Kurtosis Values of Each PF Component												
	*PF* _1_	*PF* _2_	*PF* _3_	*PF* _4_	*PF* _5_	*PF* _6_	*PF* _7_	*PF* _8_	*PF* _9_	*PF* _10_	*PF* _11_	*PF* _12_	*PF* _13_
Normal state	25.43	20.03	6.45	7.49	6.69	4.46	2.77	3.24	2.02	2.74	3.69	2.48	1.84
VPG	12.65	6.33	6.07	4.95	6.06	8.25	2.60	2.74	1.99	1.91	6.99	4.41	1.53
VPF	22.50	8.66	4.79	6.24	8.30	4.61	2.91	4.36	1.57	1.61	8.23	3.53	1.91
SFS	11.70	7.65	6.14	6.88	9.13	9.61	3.88	2.59	1.63	2.50	6.27	2.77	2.11

**Table 2 entropy-24-01480-t002:** Identification results of the four states of the valve.

Feature Extraction Method	Identification Accuracy (%) of Valve State				Total Accuracy (%)
	Normal State	VPG	VPF	SFS	
AUPLMD–RTSMWPE	95	90	95	97.5	94.375
AUPLMD–MWPE	90	87.5	90	90	89.375
LMD–RTSMWPE	92.5	90	92.5	92.5	91.875
LMD–MWPE	87.5	87.5	85	90	87.5

## Data Availability

The datasets used or analyzed during the current study are available from the corresponding author uon reasonable request.

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
