# Peer review of "Fault Diagnosis Method Based on AUPLMD and RTSMWPE for a Reciprocating Compressor Valve"

_entropy, 2022, doi:10.3390/e24101480_

Round 1
Reviewer 1 Report
A fault diagnosis method combining adaptive uniform phase local mean decomposition (AUPLMD) and refined time-shift multiscale weighted permutation entropy (RTSMWPE) is proposed for the reciprocating compressor valve. Simulation and experimental datasets validate the effectiveness and superiority of the proposed method. Some issues are presented as follows.
1. The merits and deficiencies of the existing techniques can be detailed. What's more, the novelty and contributions can be highlighted in the introduction.
2. In part 2.4 simulation signal analysis. The simulation is well designed for algorithm verification. But this is not related to the fault signal characteristic of compressors. It is better to conduct related analysis on task-specific simulation signals.
3. Different decomposition methods have been carried out. The entropy value is used for evaluation. Are there other evaluation indexes for full comparison?
4. The algorithm is implemented on the data collected under constant operating conditions. But in the real application, the operation condition of machines will change. How about the performance when the proposed method is applied to the test samples from other unknown conditions?
5. The English in the present manuscript can be improved. E.g., “As a comparison, respectively perform AUPLMD and RTSMWPE, AUPLMD and MWPE, LMD and RTSMWPE, and LMD and MWPE of feature vector diagnosis under the same data sample.”
Author Response
Please see the attachment,Thank you very much!

Reviewer 2 Report
Dear Authors,
Please find attached my comments and review.
Kind regards,

Round 2
Reviewer 1 Report
All the issues have been addressed.
Reviewer 2 Report
The paper has been improved, but it still needs to improve the figures.
All the Best.